# Training Teachers in China to Use the Philosophy for Children Approach and Its Impact on Critical Thinking Skills: A Pilot Study

**Caiwei Wu**

School of Education, Durham University, Leazes Road, Durham DH1 1TA, UK; caiwei.wu@durham.ac.uk

**Abstract:** Philosophy for Children (P4C) is an educational approach that helps children question, reason, construct arguments, and collaborate with others. This approach to teaching is new to Chinese teachers and students who have traditionally relied on rote learning and dissemination of knowledge. Independent thinking and questioning are rarely encouraged. This article reports on a pilot study aimed at training teachers in one school in mainland China to use P4C to promote thinking skills. Six year 7 classes (age 12–13) and their teachers were randomly assigned to receive P4C training ($n = 90$ pupils) or to a control group ($n = 88$). The intervention ran for 4 weeks. The study found that teachers appreciated the P4C methods but were concerned about using the method in their regular curriculum. An impact evaluation shows that students who were taught P4C experienced a small improvement in thinking skills, measured using a composite of validated critical thinking tests.

**Keywords:** philosophy for children; thinking skills; teaching pedagogy development

## 1. Introduction

The promotion of higher-order thinking skills is not new but has been increasingly highlighted as skills necessary for the 21st century, especially with the rapid growth and dissemination of information globally, not all of which is trustworthy [1]. Following the dramatic changes that have taken place in Chinese society, the target of education has shifted to all-round development of citizens. To be competitive globally in business, education, science, and technology, China has emphasised the promotion of critical thinking rather than simply imparting knowledge and fulfilling exam requirements.

In 2010, the Chinese government released the 'The National Guidelines for Medium and Long-Term Educational Reform and Development (2010–2020)', which emphasised skills such as independent inquiry, cooperation, communication, and problem solving as well as fostered cognitive skills [2]. To meet the aims of the reform, professional training was needed to help teachers access new pedagogical approaches and skills.

Although the government has provided some training, researchers have criticised it as inefficient [3,4]. They were very theoretical and consisted largely of hours of listening to government policy and regulations [3,4]. They reported that teachers complained of having to listen to theory-based lectures in large groups [4]. There was little training in practical application of the new approaches, and specific guidance for practical learning activities was lacking [5]. For example, in the training programme, teachers were simply told the importance of thinking skills and why they are beneficial for the development of children's cognitive skills but were not given any practical examples of how to develop thinking skills in students. Therefore, despite the provision of teacher training, implementing new pedagogies is still challenging for many teachers.

An effective teacher development programme should be one where teachers are exposed to the types of activities and lessons that would be performed in the classroom [6]. Similar perspectives also presented that teacher professional training should have high applicability, a practical orientation, participation, and interactivity [7]. It should be a process

that interlaces and complements theory and practice. This was not the case in China; the education reform was introduced without an accompanying effective professional development programme. It is possible that those responsible for the training were ill-equipped to teach the new curriculum, not having been exposed to critical thinking themselves.

Against this background, there is now recognition for a more practical approach to teacher training. One programme that has been considered potentially useful for promoting critical thinking and for developing higher-order thinking skills is Philosophy for Children (P4C). Philosophy for Children (P4C) is an educational approach focused on philosophical enquiry and dialogic teaching. The aim of the intervention is to help children develop their abilities to reason, question, construct arguments, and communicate collaboratively with others [1,8]. It emphasises carefully constructed dialogue between teachers and students, and among students [9]. The philosophical dimension of teacher inquiry depends largely on teachers' knowledge of philosophical issues as well as on pedagogical content knowledge [10]. Training teachers is therefore essential for successful implementation of the programme.

## 2. Background

### 2.1. Philosophy for Children

P4C was first developed by the American Philosopher Matthew Lipman in 1972. While Lipman was teaching philosophy at Columbia University, he claimed that the educational system failed to help students become independent thinkers. Under this background, Lipman envisaged programmes for teaching critical thinking and informal logic. Borrowing from Peirce and Dewey, who were initially put forward and broadened the scope of 'Community of Inquiry', Lipman systematically applied the concept of 'community of enquiry' to the educational setting and combined it with teaching thinking skills. He argues that 'classroom is a type of community of inquiry, which leads to questioning, reasoning, connecting, deliberating, challenging, developing problem-solving' [11].

Lipman also wrote novels that included various philosophical topics for students. These materials were based on discussions of philosophical issues in line with Piaget's approach to cognitive development. He hoped that students might raise questions according to the philosophical novels and might explore issues through reasoning, asking for examples, questioning hypotheses, and testing hypotheses.

P4C teaching is not a skill that teachers can easily pick up and deliver. To facilitate philosophical inquiry, teachers need to be trained both in content knowledge and pedagogical skills. This is even more necessary with Chinese teachers whose approaches to teaching are highly teacher-centred and have been primarily one of disseminating knowledge.

This philosophical approach to teaching and learning was supported by the Institute for the Advancement of Philosophy for Children (IAPC), which develops curriculur materials for schools. These materials are designed to engage children in thoughtful discussions regarding the epistemological, ethical, social, and aesthetic dimensions of philosophical experience to support them in making informed choices. According to the IAPC website, a typical P4C session would involve children reading aloud or acting out philosophical stories and ponder on issues that are relevant to their experience. The role of teachers is to facilitate these dialogues, for example, by encouraging students to share their questions and ideas with their classmates. Teachers also model examples of good thinking such as clarifying terms, giving good reasons, offering counter arguments (explanations), drawing inferences, and challenging assumptions. The social aspects of dialogue such as listening to each other and building on each other's ideas are reinforced. Teachers guide the discussion and share their perspectives about the issues [12].

In the UK, the Society for the Advancement of Philosophical Enquiry and Reflection in Education (SAPERE) was established to provide training courses since 1992. The SAPERE's training methods are different from that of IPAC. SAPERE suggests various materials, which include stories, poems, films, images, pictures, and books, rather than asks for specially written philosophical novels. Currently, SAPERE is a leading service provider,

and its sister organisation, named P4C, in China is the only organisation that can provide P4C training in mainland China.

## 2.2. The P4C Intervention

The impact of P4C on thinking skills and academic outcomes has been evaluated in several studies, most of which were conducted in the US and in the UK. The earliest evaluation by Lipman and colleagues [13] included 20 pupils from two schools in the Montclair District. The matched comparison study reported significant gains in logical reasoning and reading. However, the results are worrying because their research design cannot be considered effective. The sample was small, and the students only received intervention for a short duration [14]. A long-term study that aimed to assess cognitive skills was conducted in 2007: 177 students attended, and they were measured using the CAT test. The results indicated that the treatment students performed better than the control group two years later although the attrition rate was high (35%). In 2015, an independently evaluated large-scale efficacy trial in England [15] found positive effects on primary school children's reading and math. However, the latest published large-scale randomised control trial presented different results [16]. There is no statistical evidence to provide evidence that the treatment group performed better than the control group. Using questionnaires, they supported the conclusion that P4C can promote classroom engagement, improve the level of respect for others' opinions, and the ability to express views clearly.

In 2004, a systematic review that included ten studies was conducted. Eight studies reported a positive impact on cognitive and noncognitive skills [1]. However, it received some criticism that the employed studies were small scale and used different measurements. A more recent meta-analysis was conducted in 2018 and employed ten studies [17]. The findings showed a moderate, positive influence on cognitive outcomes and an apparent positive effect on reasoning skills. In summary, most of the research results in Western countries that use P4C as an intervention are positive, although they focused on different abilities.

P4C is not new in China. In 1997, the earliest P4C practice was developed in Yunan Province. In this project, teachers' feedback on P4C was quite positive. Assessed via a questionnaire, 89% of teachers believed that P4C could promote students' thinking and 83% of teachers believed that their dialogic skills improved. However, there were no standardised test results to prove whether students made progress [18]. In the last 10 years, the number of studies on P4C in Chinese educational contexts has increased. However, most of them have made theoretical contributions, for example, to explore the possibility of combining Confucian dialogue and the community of inquiry [19] and to analyse what types of themes and topics are suitable for the Chinese classroom [20]. In addition, some empirical studies were developed in Hong Kong and Taiwan. A study involving 28 first-year secondary students in Hong Kong showed that P4C students made bigger gains on the New Jersey Test of Reasoning Skills, with an increase of 27.9% from the pretest compared to the control group, which achieved only 13.3% improvement [21]. The treatment group was taught by the researcher, who had attended a 10-day training workshop run by IAPC. The control group was taught by an English language teacher at the same school. Since there were only two teachers, it is not possible to attribute the results to the intervention alone. The second study was a quasi-experimental study conducted in Taiwan, which randomly assigned two classes of secondary school students to P4C or not ($N = 62$) [22]. P4C was delivered as part of an after-school activity. The results showed that the control group made bigger gains between pre- and posttest than P4C students. Since students were randomised by classes ($N = 2$), the study cannot rule out class and teaching effects. The impact of P4C on Chinese students is therefore inconclusive.

*2.3. P4C Training*

With education and teaching pedagogy reform, we cannot forget to train the teachers with the new teaching skills.

A medium-scale empirical research was conducted in the UK [23] and presented the complete training process, which included how they introduced the community of enquiry, what exercises were demonstrated to improve dialogue, and what books and resources they suggested that the teachers read. Similarly, a study related to developing dialogic teaching was presented in 2018 [24]. In addition to basic theoretical training, the follow-up training service in that project showed potential as a model of learning. After the teachers received theoretical training, the training team also provided mentors to each school. They shared real video records to help teachers find similar problems within their class. They also provided guided planning, target-setting, and review. SAPERE has conducted two large-scale P4C research studies in the UK since 2015 [14,15]. They run an independent training course system that can meet the needs of different levels of difficulty. For example, in the Level 1 foundation course, they introduce the theory, provide tools to develop questioning and thinking skills, connect P4C to the curriculum, demonstrate P4C in practice, and present resources and sources for teaching materials. In the Level 2A course, they focus on facilitating P4C enquiries with more flexibility and reflection.

P4C was introduced in China nearly 30 years ago. However, most Chinese teachers do not have P4C teaching experience. The earliest P4C practice provided seven-day training to practitioners. [18]. However, the authors discussed neither what methods they used nor what types of activities were part of the training. They only mentioned general views about P4C using a questionnaire. For example, 'Is P4C suitable for our country?' and 'Can a community of inquiry improve the efficiency of teaching and learning?' Their work makes a poor reference for future research in P4C professional development. In small- and medium-scale studies, researchers are more likely to employ experienced P4C teachers rather than to train teachers who have never experienced P4C before [21,22,25,26]. Therefore, P4C professional development is a relatively weak part of the academic research field in China [27].

In summary, previous literature provides evidence that P4C has a positive impact on various abilities, especially reasoning skills. However, most of them were conducted within a Western educational background. Although researchers who focused on Chinese content have contributed to the promotion of P4C theory, the number of empirical studies is limited and there are few studies on teachers' professional development in this field.

## 3. Aims and Objectives

The aims of this pilot are as follows:

- Test the feasibility of delivering P4C lessons in a regular Chinese school;
- Evaluate the professional development model developed for training of the teachers in the delivery of P4C;
- Test the training resources (i.e., lesson plans) developed by the researcher;
- Test the measurement tools (i.e., the critical thinking test) to see how long it will take and if the questions are appropriate for the educational level of the students;
- Identify potential barriers/challenges in the staff training and classroom delivery of P4C.

## 4. Methodology

*4.1. Trial Design*

This pilot study was conducted in one state school in Fushun, Liaoning Province, in northeast China. This is an average region in terms of economic and educational development. The study design was a two-group pretest posttest randomised controlled trial. Randomisation was at the class/teacher level. Random assignment of the participants ensured that any confounding variables were equally distributed between the two groups. To avoid bias, randomisation was blinded to the participants (teachers and students) and

group allocation was revealed only after the pretest. There is research evidence suggesting that knowledge of group assignment could influence the way students perform on the test [28].

The purpose of the pretest was to establish baseline equivalence. The pretest was administered before randomisation and before the first session was delivered. A posttest was taken after the last P4C lesson. The primary outcome was students' critical thinking skills. Impact was estimated using the gain scores between the pretest and posttest, expressed as Hedges' g effect size. This is the difference in the mean gain scores between the two groups divided by the pooled standard deviation.

In addition, a process evaluation was carried out to monitor fidelity to treatment and to identify challenges or barriers faced by teachers in the implementation of P4C as well as to collect feedback on the teacher training. The process evaluation was conducted using classroom observations and interviews with the teachers.

### 4.2. Participants

The participants included six year 7 classes for a total of 178 students (aged 12–13 years) and two teachers. The teachers of three classes were randomly assigned to the experimental group ($N = 90$; 1 teacher), and the other three classes were signed to the control group ($N = 88$; 1 teacher), in which they continued regular lessons. The students in the experimental group received seven P4C sessions during the four weeks of the study.

Year 7 classes were chosen because the curriculum pressure was relatively small compared to that of year 8 and year 9 classes. The school was secular, and the school population was ethnically homogeneous (all were Chinese). The students were predominantly (more than 90%) from middle- or working-class families.

It is not standard practice to ask young children (age 11–12) for consent in this kind of research. Consents, where required, are normally sought from parents. The lessons were delivered as part of the school curriculum where consent from the school leaders have already been sought. No personal sensitive data was collected in this study, and consent from parents was mainly to ask for permission to use pupil test data.

### 4.3. Procedure

Before the trial, the pilot teacher attended two days of P4C training (Level 1) in Shanghai. The training was provided by *P4C in China*, which is SAPERE (UK)'s sister organisation in China and uses standardised SAPERE methods.

The trial started on 1 September 2019. The P4C lessons, each taking 40 min, were taught twice a week for four weeks. The lessons were taught during English class. A total of seven lessons were observed. They were recorded by handwritten field notes and video. In this way, not only can the researcher record key segments but also the video can be helpful when reviewing the dialogue. The P4C session proceeded as follows: the teacher first presented a stimulus. The stimulus could be a story, a picture, or a video. Here, we used the story of Cinderella as an example. Then, students were given individual thinking time and they were allowed to take turns sharing with their peers what they found interesting about the stimulus. The teacher helped and guided students in raising philosophical questions. After that, they chose a question that was supported by the majority and discussed it as a group. The question could be 'Should parents ask their children to do housework?' This is not a factual question. Students were allowed to present different opinions. They were encouraged to think hard about what reasons or examples could be used to support their ideas and to critique others. In contrast to a traditional Chinese classroom, the teacher worked as a facilitator rather than as an authority.

Students in the control group were taught by another teacher at the same school. To avoid contamination, the control group teacher did not teach any topics related to philosophy and thinking skills. Instead, the teacher taught the normal curriculur content using traditional methods.

Finally, both groups were administered the posttest on 29 September 2019. The teacher and students in the experimental group were interviewed informally at the end of the trial to obtain feedback regarding their experience with the P4C lessons and the challenges faced as well as to identify optimal ways to overcome these challenges.

## 5. Teacher Training

Prior to the trial, the teachers were trained in the delivery of P4C lessons. The training was provided by P4C in China (Shanghai), which is SAPERE (UK)'s sister organisation in China. SAPERE (Society for the Advancement of Philosophical Enquiry and Reflection in Education) is the UK charity supporting P4C. Its mission is to train teachers in P4C. P4C in China (Shanghai) also provides standardised SAPERE training methods. As mentioned above, traditional teacher training provided by the government is theoretical and lacks practical guidance. For this study, therefore, more detailed guidance was provided to the teacher. In addition to imparting theoretical knowledge, the trainer demonstrated the entire process of the P4C programme several times, allowing the teacher to experience how the new approach works in a real classroom situation.

The Level 1 P4C course lasted two days. Eighteen participants from different provinces attended the course. Twelve of them were kindergarten teachers, and the rest of them were primary and secondary school teachers. The training sessions ran from 9:00 to 12:00 and from 13:00 to 16:00 each day. The training included the following elements:

- Introduction of P4C, key principles and methods;
- Demonstration of P4C lessons and key practice;
- Sharing of available resources and teaching material;
- Advice and support.

### 5.1. Principles of Philosophy for Children

Generally, a P4C class has a relatively standard structure. There are five main steps in a P4C session:

- **Start the inquiry**: students and teachers sit in a circle so that everyone can easily hear each other.
- **Share a stimulus to prompt inquiry**: the American educational reformer John Dewey believed that all enquiry begins with a problematic situation. Therefore, presenting a challenging stimulus is the basis for a successful P4C class. Stimuli may include stories, poems, picture books, videos, and news articles. The purpose of the stimulus is to introduce the topic and to generate interest.
- **Encourage students to think**: this step allows time for individual thinking and public reflection. After a couple of minutes, the members of the discussion group are encouraged to share their responses to the stimulus.
- **Question and discussion**: after the students share their ideas about the stimulus, a philosophical question that can be discussed during the remainder of the session is put forward. During this time, the students listen to others; question their peers; and present their arguments with reasons, evidence, and examples. The teachers should play a supportive role. They can employ a Socratic questioning model to promote deeper discussion and to guide students' thinking.
- **Have the students evaluate, build ideas, and summarise**: this step allows students to express their final thoughts. Ideally, everyone provides an evaluation and summary of the discussion.

### 5.2. The Key Practice

To help teachers have a better understanding of how P4C works in the classroom, the trainer demonstrated and modelled a P4C lesson focused on creating questions and developing discussion.

### 5.2.1. The Stimulus

At the beginning, the participants were trained to prepare the stimuli. Picking out a stimulus with significant concepts helps encourage pupils to lead the discussion to deeper levels. During the training, the participants prepared an image as the stimulus. They shared their thoughts about what interests them or puzzles them, or what they find significant. By comparing participants' evaluation and feedback of different images, the teachers can find out what types of stimuli are suitable for the P4C class. The trainer also gave some suggestions that could be used for stimuli such as the following:

- *Literature: love, democracy, fairness, justice*
- *Humanities: justice, truth*
- *Citizenship: rights, duties, freedom, welfare*

### 5.2.2. Questioning and Dialogue

Unlike the traditional Chinese pedagogy of cramming and memorisation, P4C encourages students to think about questions on their own rather than to receive answers directly from their teacher. To achieve this goal, the students were given enough space and time. Question-creation may include talking to partners, writing questions down, and displaying questions.

In P4C lessons, dialogue is a key element of the programme. It promotes deeper engagement and a level of understanding. In a P4C class, the dialogue generally includes *asking questions, giving examples and evidence, discussing and critiquing, summarising, and evaluating*. To improve the quality of the dialogue, the Socratic method of questioning was suggested. For example, *'Do you mean . . . '* to clarify their questions and *'Do you think your question is similar or different from others'* to help summarise. The discussion could focus on providing alternative points of view, giving examples, examining reasons, and establishing logical statements. Some cards with Socratic questions were provided in the training. It could be used for both teachers and students during the discussion. Here are some examples:

*Information process questions*

- *Could you explain what you mean?*
- *Can someone give an example?*

*Reasoning Questions (expanding and probing)*

- *What are your reasons for saying that?*
- *Do we have any evidence?*

*Enquiry Questions*

- *So you agree/disagree with . . . ?*
- *Is that always the case or only sometimes?*

*Creative Thinking Questions*

- *What if. . .*
- *Does.. imply. . . ?*

*Evaluation Questions*

- *Have we reached any conclusion?*
- *Can anyone summarise what we have said so for?*

### 5.3. Sharing of Available Resources and Teaching Material

In this pilot study, the participants were given the opportunity to practice the steps and the questioning techniques. They were also provided instructional materials as a theoretical supplement. These were available in print and as online videos. The following were included:

- *A handbook provided by P4C in China that includes the aims, process, details, and P4C theoretical knowledge;*

- *A booklet containing lesson plans and transcripts of lesson extracts;*
- *Video material: videos of recorded P4C classes in a Taiwan secondary school of P4C lessons in real classroom situations from which the teachers can learn and review real classroom practices.*

*5.4. In-School Support*

In addition to the introductory P4C training, additional monitoring support was offered. First, the researcher helped the pilot teacher prepare lesson plans. The philosophical topics and stimuli were selected in consultation with the teacher. The question enquiry and discussion parts were first designed by us, with additional materials supplemented by the teacher. The researcher then observed the delivery of the P4C session each week and provided feedback on the lesson. The pilot teacher was encouraged to reflect upon each lesson.

In summary, the training course introduced the theory and practice of P4C, provided tools to develop questioning and thinking skills, demonstrated philosophical enquiry in practice, and presented P4C resources for compiling teaching materials. The training provided the teacher with the skills, knowledge, tools, and resources to teach P4C.

## 6. Data Collection

The following three instruments were used to collect data for this study.

*6.1. Thinking Skills Test*

The primary objective was to improve students' critical thinking skills. The choice of an appropriate test was a challenge as the test had to be validated and appropriate for the language ability of first-year secondary school students for whom English was not their first language. An English–Chinese edition was provided for students to help them understand the questions.

Their critical thinking skills were measured using a bespoke test made up of components from the three commonly used tests for critical thinking: CAT, the Cornell Test Level 1 [29], and the Waston Glasser test [30]. All of the tests have limitations, however. They are either too long, too expensive, or not age-appropriate. As a compromise, the researcher selected appropriate questions from these existing tests and developed a test that is suitable for the students in the study. It includes tests on inferences, assumptions, deduction, interpretation, arguments, verbal and nonverbal reasoning, spatial reasoning, quantitative reasoning, analogical reasoning, and inductive reasoning. The tests included 12 questions to be completed within 30 min. All questions were in multiple-choice format. An example of the test questions is presented below (see the pretest questions in Appendix A). As the test is a test of critical thinking and not language skills, the researcher prepared the questions in both Chinese and English.

*6.2. Classroom Observation*

In addition to collecting data from the pre- and posttests, all P4C lessons were observed. The observations were recorded as field notes by the researcher. For future training and for feedback to the teacher, all sessions were also audio-recorded. It mainly aims to document the interaction among students and the teacher during discussions in the classroom. The level of student engagement in terms of asking and answering questions is noted. It helps to determine whether students demonstrated critical thinking skills after exposure to P4C and teachers' use of the P4C protocol.

*6.3. Informal Interview*

As part of the process evaluation, nearly ten students and the pilot teachers were asked for their opinions of P4C lessons. This included questions about whether they enjoy the sessions, which part of P4C they liked or disliked, the quality of dialogue between peers and with the teacher, and what challenges they faced. The interviews were unstructured

and informal. The purpose of the informal conversation was to gather a general overview of the pupil's attitudes towards P4C.

## 7. Findings

### 7.1. Findings from the Impact Evaluation

In order to answer the research questions, it is necessary to evaluate the impact of P4C on students' critical thinking between the experimental and control groups. A significance test and its variance (e.g., confidence intervals and *p*-value) are not appropriate in this study. What the *p*-value in significance tests shows is how likely it is to obtain our results, assuming that there is no difference between groups. It does not tell whether within the given results there is a difference between the groups. Therefore, Hedges' g effect size was used to estimate the gain scores between the pretest and posttest. This is the difference in the mean gain scores between the two groups divided by the pooled standard deviation. The formula is *Effect size = (Mean treatment gain score − mean control gain score)/pooled standard deviation (average SD)*.

At the beginning, the experimental group had a slightly higher pretest score (M = 5.27) than the control group (M = 5.24). After one month of intervention, the posttest mean score of both groups improved. The experimental group showed a greater percentage increase (5%) than the control group (4%). This indicates that students who received P4C intervention made greater progress in the thinking tests than those who did not.

The overall results show that students who were taught P4C made slightly bigger gains than those in the control group who received the regular lessons (Tables 1–3). This suggests that P4C has a small positive impact on students' critical thinking skills after only one month of intervention (effect size +0.03).

**Table 1.** Descriptive statistics for pretest scores.

| Group | Pre-Score Mean | SD | ES |
|---|---|---|---|
| Experimental group (N = 86) | 5.27 | 1.29 | |
| Control (N = 87) | 5.24 | 1.34 | |
| Overall (N = 173) | 5.25 | 1.35 | 0.02 |

**Table 2.** Descriptive statistics for posttest scores.

| Group | Post-Score Mean | SD | ES |
|---|---|---|---|
| Experimental group (N = 86) | 5.57 | 1.47 | |
| Control (N = 87) | 5.49 | 1.31 | |
| Overall (N = 173) | 5.53 | 1.39 | 0.05 |

**Table 3.** Overall gain score.

| Group | Gain Score | SD | ES |
|---|---|---|---|
| Experimental group (N = 86) | 0.31 | 1.59 | |
| Control (N = 87) | 0.25 | 1.85 | |
| Overall (N = 173) | 0.28 | 1.72 | ±0.03 |

### 7.2. Classroom Observation

In addition to using standardised tests to examine the role played by P4C in developing students' thinking skills, classroom observations of theP4C lessons were used to determine the changes in classroom dialogue. In the four weeks of the intervention, it was observed that students' engagement in the classroom and the willingness to express themselves increased. The teacher gradually handed over control of the class discussions to the students. Students talked more, and the lessons became more student-oriented and less teacher-focused.

Chinese students are typically quiet and passive learners. For most of the time in class, they are more likely to receive knowledge from the teacher as an authoritative figure than to contribute to the discussions. Students' voices are rarely heard. The role of the teacher is to disseminate information, and that of the students is to receive the information. They are not encouraged to question the information imparted upon them. The traditional teacher-centred pedagogical style in China provides limited opportunities for students to express themselves. In this study, when the teacher adopted the role of a facilitator rather than as a figure of authority, the students initially found it difficult to engage. As the lessons progressed, they became more willing to participate and to contribute their ideas.

To demonstrate the progression, we selected two short dialogues between the teacher and students, one from the first week and one from the second week of the intervention. They show how the teacher was better able to probe and ask open-ended questions to get the discussion going. The students similarly show their willingness to provide more expanded answers. Other students also joined in the discussion, and they were keen to provide a reason for their responses. This was something that was never explicitly encouraged.

*Week 1*
**Teacher:** *Why do you study?*
**Student:** *Because I want to go to university.*
**Teacher:** *Is this the only reason you want to study?*
**Student:** *Erm...[keeps quiet]*

*Week 2*
**Teacher:** *Can you enjoy learning as you do gaming?*
**Student A:** No!
**Teacher:** *Why not?*
**Student A:** *I don't know. But I know I love playing computer games but I don't like studying.*
**Teacher:** *Why do you like playing games?*
**Student A:** *I get satisfaction from the game, especially when I win or complete a difficult level.*
**Teacher:** *Why can't you get satisfaction from learning?*
**Student B:** *I can. I love maths—it challenges me—I feel satisfaction when I solve difficult questions.*
**Student A:** *I don't agree with you. You're good at maths...I work hard but still can't perform well in maths. But I can win the game if I try a few more times.*

At the beginning, the students felt confused and unwilling to talk. During the second week, however, their answers became longer than in the first week. In addition, the conversation was not only between the teacher and a single student: other students also participated in the discussion. During the first two weeks, however, there was not much evidence of higher-order thinking. Changes were also observed in the teacher's questioning techniques in the third and fourth weeks, a dialogue of which is shown below:

**Teacher:** *Why do you think Tony and Bill are friends? Can you give me some reasons?*
**Student A:** *Because they play together.*
**Teacher:** *Do you mean people who play together are friends?*
**Student A:** *Erm...no.*
**Teacher:** *Oh? How do you define 'friends'?*
**Student A:** *I don't know.*

This was an example during the fourth week when the teacher applied Socratic questioning more frequently than before. The questions she put forward were more effective in facilitating thinking. For example, 'Do you mean ... [repeat student's point]?' helps students find contradictory points in their answers. Asking students to give a definition is also a good way to practise summarising. In the discussion, the students also displayed their thinking processes, such as identifying contradictions, giving counter examples, and summarising.

**Student B:** *No, friends can play together, but I can't be sure that people who play together are definitely friends.*
**Student C:** *I agree. I don't think Tony and Bill are friends.*
**Teacher:** *Why?*
**Student C:** *Although Tony plays with Bill, Tony is unhappy.*
**Teacher:** *Why is Tony unhappy?*
**Student C:** *Because Bill broke Tony's new bike and his toys—he doesn't respect Tony.*

Similarly, during this example, when the students were challenged to give deeper answers, the teacher asked 'Why?' twice rather than giving the answers directly. In a traditional Chinese classroom, students expect the teacher to give them the answer, and the teacher is often quick to offer the answer as they think that this is the role of the teacher. In the four weeks of intervention, the teacher was observed to consciously become more of a facilitator. She tried to reduce her control of the classroom and gave students more opportunities to think independently.

Overall, throughout the four weeks of P4C intervention, the teacher and students made progress. The students changed from passive receptors of knowledge to active participants in classroom discussions. They started to engage in more logical and reasonable dialogue rather than simply providing 'yes' or 'no' answers. Moreover, the teacher was more likely to apply Socratic questioning techniques and to put forward high-quality questions as a facilitator in the class.

*7.3. Interview Findings*

In addition to collecting data from the thinking test results and classroom observation, informal interviews were conducted to capture both the teacher's and students' general feedback towards the P4C intervention. The primary target of the pilot study was to test the feasibility of the training model and to identify potential barriers that may challenge our study in the future. Therefore, the interview questions that were prepared for the teacher revolved around the following themes: What is your impression of the training? What do you like or dislike about P4C? Are there any challenges to applying P4C in your classroom? The students were asked "Do you like P4C lessons?"

Firstly, the pilot study teacher expressed her appreciation of the P4C training as follows:
*I think the training is good. It not only introduced the principles of the P4C approach, but a large part of the training included classroom simulations. The trainer worked as a facilitator modelling the delivery of P4C, while I acted as a student. This allowed me to see how the P4C pedagogy works in practice. It also helped me to think about the actual situation and the challenges we may encounter in the classroom.*

However, the teacher was concerned about how to implement P4C into regular Chinese classes and whether the P4C pedagogy can be generalised and incorporated into other subjects within the curriculum. Although the training provided a few ideas on how to go about doing this, the teacher commented that: *I used a specific time each Friday to deliver P4C in place of the regular lessons, but after the project was completed, I would have to revert to the national curriculum and its lesson plans. Thus, it may be challenging for us to integrate the P4C pedagogy into the regular curriculum content.*

When the researcher asked the teacher what her impression of the P4C pedagogy was, she provided positive feedback: *Within the span of one month, I think the most positive impact was classroom engagement. I can see that the students' participation has greatly increased during this time. There was a clear difference between their active performance in my P4C class and their tendency to remain silent in the past. And I think that is a great beginning for facilitating thinking because if they talk more, they will think more.*

For the student interviews, the researcher randomly selected five students and invited them to share their opinions. Three key words that were prominent among the students' feedback were 'relaxed', 'respectful', and 'learn from others'. Here are some of their comments:

- *I like the P4C class because I have a lot of amazing idea to share with my classmates.*
- *I'm very happy because the topics are interesting in the P4C class. It also makes me relaxed because the teacher does not judge me at the moment.*
- *I progressed a lot. I learned how to present my ideas and theories, and how to use evidence to support them.*
- *I like hearing how other classmates think and talk. I am curious about other people's mind.*

However, while the students enjoyed the new programme and teaching style and appreciated the opportunity to learn from one another and to exchange opinions, some students doubted whether P4C classes could help them reach their academic goals. As one student expressed: *I don't understand what the relationship between the P4C lessons and exam requirements is.*

In summary, both the teacher and students expressed their critical attitudes towards P4C intervention. The teacher agreed that the P4C training was helpful to her teaching skills, but she also hopes to receive guidance on how to integrate it into subject teaching. Moreover, while the P4C approach increased the level of energy and engagement in the classroom because it provided students with opportunities to share their thoughts, the students doubted whether it is ultimately beneficial to their academic performance.

## 8. Discussion

This pilot study reports an attempt to incorporate P4C in a Chinese secondary school classroom. The outcomes are positive. The pilot teacher was able to complete the P4C lessons, and the students gave positive feedback on the new teaching methods. However, these results should not be accepted without criticism.

### 8.1. P4C Training

In this pilot study, the P4C organisation in China provided the training, which followed the SAPERE method. Few previous studies revealed the details of their P4C training. Gao [27] also suggested that there should be more research focusing on the professional development of training in P4C. That author introduced their training content and activities. It included how to select stimuli, how to facilitate questioning and dialogue skills, and what materials were provided to the teacher. However, the two-day P4C training is not sufficient for teachers who do not have experience with P4C. It is necessary to provide follow-up training as a supplement.

One possibility is a monthly seminar with a P4C trainer. Professional trainers can help teachers strengthen their theoretical knowledge and can increase the depth of content. Additionally, they can address the problems that teachers face in practice from the previous stage. Another possibility is to invite the experimental group teachers for peer discussions. There should be opportunities and discussion forums for teachers to exchange experiences, ideas, and challenges. They may encounter similar problems in practice. Peer discussions allow for sharing solutions with each other.

### 8.2. The Application of P4C

The trial lasted for a month. Both the teacher and students gave positive feedback on P4C. The teacher learned a new teaching pedagogy and improved her dialogic skills; the students improved their thinking and expression. However, some barriers were found.

The first barrier is the design of the P4C lesson. So far, there is no P4C textbook that is specially focused on Chinese content. Most materials were translated directly from the textbooks of Lipman and IPAC, which were created based on Western educational background [12,31]. This may lead teachers to think that the content of P4C is not related to Chinese curriculum requirements and is thus not suitable or helpful for Chinese students. In this study, the training provided skills about how to design module and materials provided rich P4C topics to the teacher. The researcher and the teacher strove to develop the localization of the P4C programme by choosing topics from the curriculum and designing lesson plans on their own. However, it is not ideal to create the lesson plan independently.

For example, due to traditional teaching habits, the teacher was more likely to choose stimuli relating to factual knowledge than controversial topics. From the feedback of the interview, students agreed that the modules are helpful for their talking and thinking, while the teacher hoped to be closer to the textbook.

To solve these problems, more materials are needed. On the one hand, it is necessary to provide more lesson plan templates, especially to present what stimuli are appropriate for P4C lessons. On the other hand, the materials need more integration with Chinese curriculur content, Chinese teachers' teaching habits, and the interests of Chinese students.

Another challenge is the application of classroom dialogic skills. Philosophy for Children is a new pedagogy completely opposed to the traditional Chinese authoritative style of teaching. It is not easy for teachers to abandon the role of authority, to change their dialogue habits in the classroom, and to create an open learning environment. In China, thus far, there are few instructions available on how teacher's dialogue skills can be effectively promoted [32]. To improve teachers' dialogue skills, it is insufficient to only learn theory—the knowledge and techniques that one acquires must be effectively put into practice in the classroom.

Therefore, helping teachers set short-term goals and providing targeted training in dialogue skills are necessary. Any reform will need to be instituted in stages. In practice, for example, the first step may be to ask teachers to move slowly from a teacher-centred to a student-centred approach and to then introduce Socratic questioning techniques to encourage students to engage in class. Teachers could then be gradually encouraged to develop high-quality discussions, giving students the opportunity to apply more higher-order thinking skills.

*8.3. Limitations and Future Research*

Although the results showed a slight improvement in the experimental group, due to the time being too short and the sample being small, our conclusions are not sufficiently convincing.

This pilot trial lasted only four weeks. Evidence from other studies shows that students who received 24 weeks of P4C intervention performed better than those in the 10 week intervention group [23]. Therefore, four weeks may be too short of a period for the impact to be realised.

The sample size was relatively small as well. The trial involved only two teachers. Any differences between the two teachers could therefore have accounted for the difference in results between the experimental and control groups. The results cannot therefore be generalised. For future work, to increase the generalisability of the results, the sample size should be increased and the duration of the intervention should be extended.

Most importantly, we suggest that academic performance should be assessed in future research because exam scores are irreplaceable to China's education system. If these P4C modifications result in greater academic achievement, it is more likely that P4C will be more widely promoted.

## 9. Conclusions

The results of this study achieved the research aims and objectives posed in the Introduction. This study demonstrated that it is feasible to train and deliver P4C lessons in Chinese classrooms, but to embed P4C in the curriculum is a challenge. Unlike previous studies that focused more on kindergartens and primary schools, this study indicates that students in secondary school are willing to accept this new teaching pedagogy.

In this study, the pilot teacher received training from P4C in China and used standardised SAPRER methods. Moreover, the training content included essential theoretical knowledge, opportunities for practice, and some useful resources. As a result, the pilot teacher positively affirmed the training outcomes.

In terms of testing the measurement tools, the modified English–Chinese version of the Critical Thinking Test was found to be appropriate to the language ability and age of

the students. They were able to understand the meaning of the questions and to complete the test within the specified time. The test results showed that the experimental group made some improvements in their critical thinking compared to the control group.

**Funding:** This research received no external funding.

**Institutional Review Board Statement:** No applicable.

**Informed Consent Statement:** Informed consent was obtained from all participants involved in the study.

**Data Availability Statement:** Data sharing not applicable.

**Conflicts of Interest:** The author declares no conflict of interest.

**Appendix A. Sample Questions in the Pre-Test for Critical Thinking Assessment**
**Pretest (Pilot Study)**

- Class
- Name
- Gender

Which one of the following conclusions is definitely true based on the statement?
1. All birds are animals and all chickens are birds

A.　All chickens are animals
B.　No chickens are animals
C.　Some chickens are animals
D.　Some chickens are not animals
E.　No valid conclusion

2. No dogs are pets. Some pets are cats

A.　All cats are dogs
B.　No cats are dogs
C.　Some cats are dogs
D.　Some cats are not dogs
E.　No valid conclusion

3. Xiaofei says he rides a bicycle every day. One day you went to his house and you saw some bicycles with flat tyres in his garden. When you see this, you

A.　You know that Xiaofei rides his bicycle every day
B.　You do not know if Xiaofei rides a bicycle every day
C.　You know that all the bicycles in his garden are Xiaofei's

4. Reading the passage below and answer the questions
4a. A group of explorers had gone to a village, called Nicoma, and disappeared. You took a group of soldiers to find out what had happened to them. You found some stone huts put up by the first group. You went into the first hut and everything was covered by a layer of dust. You called out but nobody answered. One of your members said: 'Maybe they are all dead.' Do you agree with his conclusion?

A.　Yes
B.　No
C.　There is not enough information to decide

4b. You send two of your soldiers to explore the area and check if the water is safe to drink. The soldier A looked at the stream by the village and reported, 'The water looks clear, it is therefore safe to drink.' Soldier B said, 'We can't tell yet if the water is safe to drink'. Which soldier is more believable?

A.　Soldier A
B.　Soldier B
C.　Neither

5. Children who go to private schools do better in exams than children who go to public schools. Mimi goes to a public school but her friend Caicai goes to a private school. This means that

A.    Mimi will do better than Caicai in exams
B.    Caicai will not do well in exams
C.    Mimi might do better than Caicai in exams

6. These words are similar in some way. Decide how they are the same. Then choose the answer choices that goes with the example word. Hen: Egg

A.    Dog: Bark
B.    Cow: Milk
C.    Peacock: Feathers
D.    Swan: White

7a. Work out which of the six cubes can be made from the left figure

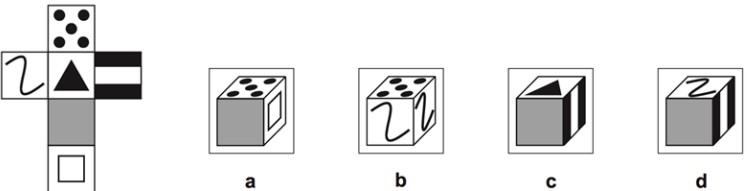

7b. Work out which option would look like the figure on the left if it was reflected over the line

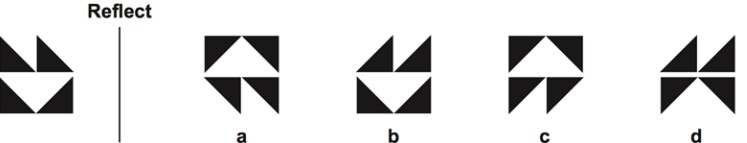

8. In each of these questions, these figures are similar in some way. Decide how they are the same and then choose the figure from the answer choices that goes with them.

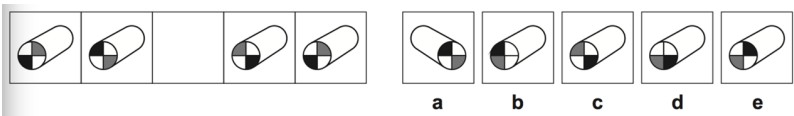

9. Find the figure in the row that is most unlike to the other figure.

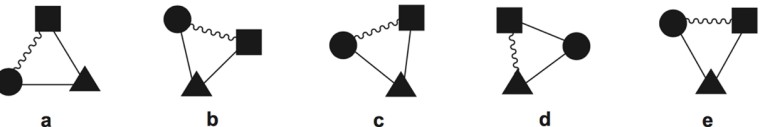

10. Lili, Wang Lei and John all walk to school each day. Lili leaves home at 8:30 a.m. and takes twenty minutes to reach school. John arrives at school five minutes after Mark, who arrives at school five minutes after Lili. Mark takes 5 min to get to school. Using this information decide which of the following statements is true.

A.    Wang Lei leaves home at 8:45

B.   Wang Lei leaves home at 8:50
C.   Wang Lei arrives at the same time as John
D.   LiLi arrives at school last

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
