# Peer review of "Training Teachers in China to Use the Philosophy for Children Approach and Its Impact on Critical Thinking Skills: A Pilot Study"

_education, doi:10.3390/educsci11050206_

Round 1
Reviewer 1 Report
Thank you very much for allowing me to read your work at this stage. I am well-aware of the developments in PwC in China and recognise the challenges and opportunities available to you in advancing this valuable work.
The work described is important in terms of moving forward practical philosophy with children in China. However, I think that the author has tried to do too much in one article and, as a consequence, it lacks the depth and impact it might otherwise have. Indeed, I can see that there are three possible articles: 1) the introduction of the training for teachers; 2) the impact on pupils’ critical thinking; and 3) a discussion of the evolution of the pupils’ dialogues.
The context of Chinese educational reform is important to the reader. While the article draws on some background literature, I would suggest that a much stronger discussion of the literature on teachers’ professional learning would support the article. This would then allow the author to focus on the training, as described. It is worth noting that SAPERE’s approach to Philosophy with Children (PwC) is not the same as Matthew Lipman’s Philosophy for Children (P4C). There are several approaches to PwC, with SAPERE’s being only one. There are others that don’t require the purchase of materials. I think this is important to the discussion, particularly in light of comments about the relevance of the materials in the Chinese context and that teachers had difficulty in embedding PwC into their practice.
It’s interesting that the author draws on a relatively recent report by the Education Endowment Fund that appears to support the case for increased attainment in children’s reading and maths. It is essential to note that their follow-up study has been published in the last week and they now suggest that the gains in attainment are not the case. Of course, there are flaws in their study, but it’s worth considering.
Assuming the author wishes to consider pupils’ thinking skills in this article or, for that matter, a new article, much more needs to be discussed in relation to critical thinking. There are studies in PwC on children’s critical thinking, but the broader literature in this area also requires some discussion as it is not at all clear what the author means by critical thinking or how it is understood in the Chinese curriculum.
Explaining how the PwC sessions worked is helpful, as is the description of the training sessions for the teachers. It is worth commenting that SAPERE level one training is very basic, and I would be surprised if, after this training the teachers were able to function well in leading pupils’ philosophical dialogue, particularly as the approach is new to them. I am also very surprised that the PwC sessions were held during English lessons. Not only is discussing ideas philosophically challenging in one’s home language, another level of complexity is added if participants are expected to do this in an additional language. This is not mentioned in the article, but it could be significant for the pupils and the teachers.
The methods section is very limited and requires much more discussion. Given that one focus was on teachers’ learning, I am surprised that no data is presented from the qualitative interview. Indeed, much more could have been made from the qualitative work undertaken, and it should reach beyond whether participants and the teacher(s) simply liked or enjoyed the sessions. With a focus on teacher learning, there is much to discuss – and evidence – in relation to the impact on practice in terms of pedagogy but also in response to the challenge of fitting it within the curriculum. Observations are also useful, but these are not discussed in the methods section. More transparency is required around what was observed, why it was observed and how observations were recorded. Similarly, I have no sense of how data was collected from the children or what data emerged. The extracts from the dialogues are interesting, and certainly merit discussion. Like I say above, this could form a separate paper, that might focus on the pupils’ contributions and also on the teacher who seems, in the shared extracts, to be very interventionist in her facilitation.
I can’t determine how results were analysed, so this needs to be clearer.
The children ought to have been asked for their ethical permission as well as their parents prior to participation in the study.
It is important to note that participation in dialogue is not the same as evidencing critical thinking.
As the author notes in the conclusion, four weeks is really not long enough time to gauge the impact on pupils’ thinking skills. This is another reason I would suggest that this element is removed from the present article and is worked into another article where more time is spent on the intervention.
In the conclusion the author notes that ‘The outcomes indicate that teachers, parents and students in the secondary school are willing to accept this new teaching pedagogy’. However, there is no evidence to support this claim. The evidence must be shared much more explicitly. It is only at this point in the article that I find that parents have been involved in the study, though no mention is made of them prior to this point aside from granting ethical permission for the children to participate in the study.
The article is written clearly in terms of its style and English language. There is an error throughout the work that is common to those whose first language is not English, and that is that often the definite article (the) is missing. There are also a couple of apostrophes that need to be fixed.
Overall, the author has something that deserves to be shared, but the article needs to be reworked. As noted above, I would suggest that it would work better as three separate articles, that background literature needs to be discussed more fully, that methods need to be clearer and that data and its analysis should be shared more explicitly. A more qualitative approach would likely strengthen much of the work. I hope my feedback is helpful in revising this work.
Author Response
I am very grateful to the reviewer for their very constructive comments and careful reading of the manuscript. I have taken on board all their comments and feedback, and addressed them in the attachment below. Thanks

Reviewer 2 Report
This is a good paper which opens up a new topic. Its method is generally correct. I would like to see citations of a few more papers from the last 10 years. I would also like to see a small explanation of the statistics used (e.g. I am assuming the differences were not statistically significant, but it is usual in papers of this kind to explain what stats you used, and state that the differences were not (or were) significant).
Author Response

(The authors gave the same response as above.)

Reviewer 3 Report
The paper presents a new topic, of interest and which can be presented and promoted to those interested in this educational form. However, the paper needs some additions to be published. It is necessary that the authors develop the portfolio of references used, which is quite small for the topic addressed.
The theoretical background can be improved, in order to show more sources specific to the approached topic. We recommend the inclusion in the list of references and other authors who have stood out in this field, through published books or programs developed for children, one being Oscar Brenifier, or books by Jostein Gaarder, Giuseppe Ferraro, Fernando Savater, and some details about Matthew Lipman program.
Avoid the use of text and related web references, sometimes in parentheses (eg Pg.5).
The information contained in the photograph presented by the authors as figure 1 are presented in detail by the authors in the text of the section, and by adding the figure the information becomes repetitive, and we recommend its presentation as an annex of the paper, just for example (with a reference to their annex)
The discussion section (8) could also contain references and interpretations of the authors on the data presented in the tables in the findings section (7), which can generate the conclusion of the paper.
The paper requires a separate section of conclusions, which is now included in section 8 Discussion.
Author Response

(The authors gave the same response as above.)

Round 2
Reviewer 1 Report
Thank you for your attention to some of the comments I previously provided. At this stage, however, the proposed article seems to focus on the aim of the study rather than the study itself. I really look forward to reading about the study as it should yield some interesting findings.
Author Response
Thank you for your further suggestions. I added some contents related to research questions in the discussion, and restructured barriers and potential solutions in the discussion. Thanks
Discussion
This pilot study reports an attempt of P4C in the Chinese secondary classroom.
The outcomes are positive. The pilot teacher was able to complete the P4C lessons and the students gave positive feedback on the new teaching methods. However, these results should not be accepted uncritically.
8.1 P4C training
In this pilot study, P4C in China provided the training service which followed the SAPERE method. In previous studies, few of them revealed the details of P4C training. Gao (27) also suggested that there should be more research focusing on the professional development of training in the P4C area. In this study, the author introduced the training content and activities. It includes how to select stimulus, facilitate questioning and dialogue skills, and what materials were provided to the teacher. However, the two-day P4C training is not enough for teachers who do not have P4C experience before. It is necessary to provide follow-up training as a supplement.
One possibility is a monthly seminar with P4C trainer. Professional trainers can help teachers strengthen theoretical knowledge and increase the depth of content. Also, to discuss the problems that teachers face in practice in the previous stage. Another possibility is to invite experimental group teachers for peer discussions. There should be opportunities and discussion forums for teachers to exchange experiences, ideas, and challenges. They may encounter similar problems in practice. Peer discussions allowed to share their solutions with each other.
8.2 The application of P4C
The trial lasted for a month. Both the teacher and students gave positive feedback on P4C. The teacher learned new teaching pedagogy and improved her dialogic skills; students improved their thinking and expression. However, some barriers were found.
The first barrier is the design of the P4C lesson. So far, there is no P4C textbook that is specially focused on Chinese content. Most materials were translated directly from the textbooks of Lipman and IPAC, which were created based on Western educational background. [31; 32] This may lead teachers to think that the content of P4C is not related to Chinese curriculum requirements, and is thus not suitable or helpful for Chinese students. In this study, researchers and teachers strove to develop the localization of the P4C programme by choosing topics from the curriculum and designing lesson plans on their own. However, it is not ideal to create the lesson plan independently. For example, due to traditional teaching habits, the teacher was more likely to choose stimuli relating to factual knowledge than controversial topics.
To solve these problems, more materials are needed. On the one hand, it is necessary to provide more lesson plan templates, especially to present what stimulus is appropriate for P4C lessons. On the other hand, the materials need more integrated with the content of Chinese curriculum content, Chinese teachers’ teaching habits and the interest of Chinese students.
Another challenge is the application of classroom dialogic skills. Philosophy for Children is a new pedagogy completely opposed to the traditional Chinese authoritative style of teaching. It is not easy for teachers to abandon the role of authority, change the dialogue habits in the classroom, and create an open learning environment. In China, so far, there are few instructions available on how teacher’s dialogue skills can be effectively promoted. [33] To improve teachers’ dialogue skills, it is insufficient to only learn theory- the knowledge and techniques that one acquires must be effectively put into practice in the classroom.
Therefore, to help teachers set short-term goals and give targeted training in dialogue skills is necessary. Any reform will need to be instituted in stages. In practice, for example, the first step may be to ask teachers to move slowly from a teacher-centred to a student-centred approach and then introduce Socratic questioning techniques to encourage students to engage in class. Teachers could then be gradually encouraged to develop high-quality discussions, giving students the opportunity to apply more higher-order thinking skills.
8.3 Limitations and future research
Although the results showed a slight improvement in the experimental group, due to the time was too short and the sample was small, the conclusion is not convincing enough.
This pilot trial lasted only 4 weeks. The evidence reflected that students who received 24 weeks P4C intervention performed well than those in the 10 weeks intervention group. [23] Therefore, 4 weeks may be too short a period for the impact to be realised.
The sample size was a relatively small as well. The trial involved only two teachers. Any differences between the two teachers could therefore have accounted for the difference in results between the experimental and control groups. The results cannot therefore be generalized. For future work, to increase the generalizability of the results, the main study would increase the sample size and extend the duration of the intervention.
Most importantly, the researcher suggested that it is significant to combine academic performance in future research, because the status of exams is irreplaceable in China’s education system. If these modifications result in greater academic achievement, it is more likely that P4C will be promoted more widely.
- Conclusion
The results of the study achieved research aims and objectives that posed at the beginning. This study has demonstrated that it is feasible to train and deliver P4C lessons in Chinese classrooms, to embed P4C in the curriculum is a challenge. Unlike previous studies that focused more on kindergartens and primary schools, this study indicates that students in secondary school are willing to accept this new teaching pedagogy.
In this study, the pilot teacher received training from P4C in China, which used standardized SAPRER methods. Moreover, the training content included essential theoretical knowledge, opportunities to practice, and some useful resources. As a result, the pilot teacher positively affirmed the training outcomes.
In terms of testing the measurement tools, the modified English- Chinese version of the Critical Thinking Test was found to be appropriate to the language ability and age of the students. They were able to understand the meaning of the questions and complete the test within the specified time. The test results showed that the experimental group had made some improvements in their critical thinking compared to the control group.
Reviewer 3 Report
(reviewed)
Author Response
Thank you for your further reply. I added some contents related to research questions in the discussion, and restructured barriers and potential solutions in the discussion. Thanks
Discussion
This pilot study reports an attempt of P4C in the Chinese secondary classroom.
The outcomes are positive. The pilot teacher was able to complete the P4C lessons and the students gave positive feedback on the new teaching methods. However, these results should not be accepted uncritically.
8.1 P4C training
In this pilot study, P4C in China provided the training service which followed the SAPERE method. In previous studies, few of them revealed the details of P4C training. Gao (27) also suggested that there should be more research focusing on the professional development of training in the P4C area. In this study, the author introduced the training content and activities. It includes how to select stimulus, facilitate questioning and dialogue skills, and what materials were provided to the teacher. However, the two-day P4C training is not enough for teachers who do not have P4C experience before. It is necessary to provide follow-up training as a supplement.
One possibility is a monthly seminar with P4C trainer. Professional trainers can help teachers strengthen theoretical knowledge and increase the depth of content. Also, to discuss the problems that teachers face in practice in the previous stage. Another possibility is to invite experimental group teachers for peer discussions. There should be opportunities and discussion forums for teachers to exchange experiences, ideas, and challenges. They may encounter similar problems in practice. Peer discussions allowed to share their solutions with each other.
8.2 The application of P4C
The trial lasted for a month. Both the teacher and students gave positive feedback on P4C. The teacher learned new teaching pedagogy and improved her dialogic skills; students improved their thinking and expression. However, some barriers were found.
The first barrier is the design of the P4C lesson. So far, there is no P4C textbook that is specially focused on Chinese content. Most materials were translated directly from the textbooks of Lipman and IPAC, which were created based on Western educational background. [31; 32] This may lead teachers to think that the content of P4C is not related to Chinese curriculum requirements, and is thus not suitable or helpful for Chinese students. In this study, researchers and teachers strove to develop the localization of the P4C programme by choosing topics from the curriculum and designing lesson plans on their own. However, it is not ideal to create the lesson plan independently. For example, due to traditional teaching habits, the teacher was more likely to choose stimuli relating to factual knowledge than controversial topics.
To solve these problems, more materials are needed. On the one hand, it is necessary to provide more lesson plan templates, especially to present what stimulus is appropriate for P4C lessons. On the other hand, the materials need more integrated with the content of Chinese curriculum content, Chinese teachers’ teaching habits and the interest of Chinese students.
Another challenge is the application of classroom dialogic skills. Philosophy for Children is a new pedagogy completely opposed to the traditional Chinese authoritative style of teaching. It is not easy for teachers to abandon the role of authority, change the dialogue habits in the classroom, and create an open learning environment. In China, so far, there are few instructions available on how teacher’s dialogue skills can be effectively promoted. [33] To improve teachers’ dialogue skills, it is insufficient to only learn theory- the knowledge and techniques that one acquires must be effectively put into practice in the classroom.
Therefore, to help teachers set short-term goals and give targeted training in dialogue skills is necessary. Any reform will need to be instituted in stages. In practice, for example, the first step may be to ask teachers to move slowly from a teacher-centred to a student-centred approach and then introduce Socratic questioning techniques to encourage students to engage in class. Teachers could then be gradually encouraged to develop high-quality discussions, giving students the opportunity to apply more higher-order thinking skills.
8.3 Limitations and future research
Although the results showed a slight improvement in the experimental group, due to the time was too short and the sample was small, the conclusion is not convincing enough.
This pilot trial lasted only 4 weeks. The evidence reflected that students who received 24 weeks P4C intervention performed well than those in the 10 weeks intervention group. [23] Therefore, 4 weeks may be too short a period for the impact to be realised.
The sample size was a relatively small as well. The trial involved only two teachers. Any differences between the two teachers could therefore have accounted for the difference in results between the experimental and control groups. The results cannot therefore be generalized. For future work, to increase the generalizability of the results, the main study would increase the sample size and extend the duration of the intervention.
Most importantly, the researcher suggested that it is significant to combine academic performance in future research, because the status of exams is irreplaceable in China’s education system. If these modifications result in greater academic achievement, it is more likely that P4C will be promoted more widely.
- Conclusion
The results of the study achieved research aims and objectives that posed at the beginning. This study has demonstrated that it is feasible to train and deliver P4C lessons in Chinese classrooms, to embed P4C in the curriculum is a challenge. Unlike previous studies that focused more on kindergartens and primary schools, this study indicates that students in secondary school are willing to accept this new teaching pedagogy.
In this study, the pilot teacher received training from P4C in China, which used standardized SAPRER methods. Moreover, the training content included essential theoretical knowledge, opportunities to practice, and some useful resources. As a result, the pilot teacher positively affirmed the training outcomes.
In terms of testing the measurement tools, the modified English- Chinese version of the Critical Thinking Test was found to be appropriate to the language ability and age of the students. They were able to understand the meaning of the questions and complete the test within the specified time. The test results showed that the experimental group had made some improvements in their critical thinking compared to the control group.
Round 3
Reviewer 1 Report
Thank you for considering the comments I sent in my previous review of the work.